# Audio Injection Adversarial Example Attack

**Xiaolei Liu** [1]  **Xingshu Chen** [2]  **Mingyong Yin** [1]  **Yulong Wang** [1,2]  **Teng Hu** [1]  **Kangyi Ding** [3]

## Abstract

We study the problem of audio adversarial example attacks with sparse perturbations. Compared with image adversarial example attacks, attacking audio is more challenging because the audio structure is more complex and the perturbation is difficult to conceal. To overcome this challenge, we propose an audio injection adversarial example attack, which provides a new sight light to increase the concealment of attack behavior. Experiments demonstrate that the proposed audio injection adversarial example attack can significantly reduce the perturbation proportion and achieve a better attack effect than traditional attack methods.

## 1. Introduction

Studies show that existing deep learning algorithms are vulnerable to adversarial example attacks especially in image domain(Szegedy et al., 2013; Goodfellow et al., 2014; Carlini & Wagner, 2017). To date, some researchers have generated audio adversarial examples successfully(Liu et al., 2020; Carlini & Wagner, 2018; Qin et al., 2019; Yuan et al., 2018). Nevertheless, the attacking purposes of all these existing methods are making automatic speech recognition(ASR) transfer sentence from '*A*' to '*B*'. The new sentence usually does not contain the original semantics. Considering the actual audio attack scenario, these traditional audio adversarial attacks are easy to be detected. To overcome this bottleneck, we propose an audio injection example attack with sparse perturbations. Our method has the following characteristics:

- **Better concealment.** Motivated by SQL injection attack(Halfond et al., 2006), this method can inject audio and append new attack semantics while preserving the original semantics. In this case, after ASR receives the audio adversarial examples, it will execute the original instruction while executing the attack instruction. At this time, the victim can still get his desired instruction execution result, which makes this adversarial example attack more covert and more in line with the actual scenario.

- **Fewer perturbations.** Unlike traditional methods of constructing audio adversarial examples, on the one hand, our method selects only a very short "silent" region at the beginning or end of the audio for injection attacks, and on the other hand, even in this very short "silent" region, we are still able to construct audio adversarial examples by adding sparse perturbations. This allows us to successfully construct audio adversarial examples by adding few perturbations.

## 2. Related Work

Deep learning algorithms have been widely applied to various applications and have shown excellent performance, however, it has been shown that these algorithms are vulnerable to small and human-invisible perturbations, which is called adversarial example attack(Goodfellow et al., 2014). Deep learning models will output incorrect results with high confidence when inputted adversarial examples.

Adversarial examples in the image domain have been extensively studied. Compared with image, audio adversarial examples are more difficult to generate owing to the Mel-Frequency Cepstrum(MFC) transform pretreatment process of the ASR system(Taori et al., 2018). Prior researches studied untargeted audio adversarial examples under white-box conditions where attackers have access to all the parameters of a network(Kereliuk et al., 2015; Gong & Poellabauer, 2017).The first study of targeted attack generated targeted audio adversarial examples for phonetic similar phrases(Cisse et al., 2017). Carlini *et al.* proved an end-to-end attack method which achieved optimization through MFC pre-processing transformation using an optimization-based algorithm(Carlini & Wagner, 2018). Some researches further studied audio adversarial example attack under black-box constraints(Carlini et al., 2016). Alzantot *et al.* Used a genetic algorithm to iteratively apply

[1]Institute of Computer Application, China Academy of Engineering Physics, Mianyang, China [2]School of Cyber Science and Engineering, Sichuan University, Chengdu, China [3]Institute for Cyber Security, University of Electronic Science and Technology of China, Chengdu, China. Correspondence to: Xiaolei Liu <liuxiaolei@caep.cn>.

*Accepted by the ICML 2021 workshop on A Blessing in Disguise: The Prospects and Perils of Adversarial Machine Learning.* Copyright 2021 by the author(s).

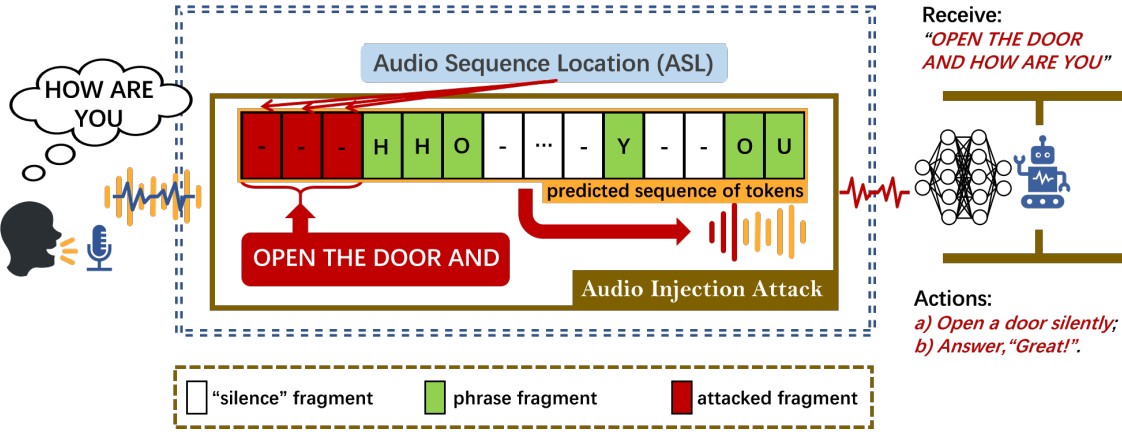

*Figure 1.* Overview of audio injection adversarial example attack.

noise to audio samples to achieve similarity and adversary at the same time(Alzantot et al., 2018). Then, Taori *et al.* expanded the genetic algorithm with gradient estimation approach on the DeepSpeech system(Taori et al., 2018).

More recent works mainly focus on the robustness and generation efficiency of audio adversarial examples. Liu *et al.* proposed two methods named Weighted Perturbation Technology (WPT) and Sampling Perturbation Technology (SPT), which improved the generation efficiency and reduced the perturbation ratio of audio adversarial examples by adding perturbations sparsely(Liu et al., 2020). Abdoli *et al.* demonstrated the existence of transferable audio adversarial examples which can fool multiple models using a penalty method to minimize an appropriate objective function on a batch of samples(Abdoli et al., 2019). Xie *et al.* proposed a universal audio adversarial perturbations generation method based on generative model. The well-trained model can generate audio adversarial examples quickly enough for the needs of real-time attack(Xie et al., 2020).

## 3. Audio Injection Attack

Audio injection attack focuses on the concealment of the attack behavior and is different from the traditional audio adversarial attack idea which completely replacing the original phrase with the targeted phrase. The overview of audio injection adversarial example attack is shown in Fig.1.

To validate its feasibility, we tried to test the common intelligent personal assistants on the market. Take Google Voice Assistant as an example, when we talk with it using the sentences in Table 1 respectively, the results show that Google Assistant successfully executed all the commands. This means when Google Assistant receives two commands at the same time, it can silently execute one command and reply to the other. Since this attack is similar to the SQL

injection attack in which additional attack commands are added after the original SQL executes the command, we call this attack an audio injection attack.

Since one second of audio can transcribe up to 50 characters, theoretically audio injection attacks have enough space to inject in most of the audio. And the length of the modified part in this adversarial audio almost negligible compared to the entire length of the audio. Although we did not actually carry out a real attack on the Google Assistant, we hope that this approach will give researchers more inspiration, that is, it is very important to preserve the original semantic information in the audio, which can increase the concealment of the attack behavior.

We use $x$ denote an audio vector, $p$ denote a phrase which is the semantic information of $x$, $y$ denote the probability distribution of $x$ decoded to $p$ and $\delta$ is the distortion to the original audio. Audio adversarial example attacks are defined as by adding some perturbations $\delta$, ASR recognizes $x+\delta$ as specified malicious texts $t$ (formally: $f(x+\delta) = t$), while there is no perceivable difference for humans. The process of generating adversarial examples can be regarded as a process of updating $x$ using gradient descent on a predefined loss function $\ell(\cdot)$ shown in Eq. 1. The iterative process stops until the adversarial example meets our evaluation requirements. *Audio Sequence Location(ASL)* is a model to help us measure the importance of each frame in the audio towards generating the adversarial examples(Liu et al., 2020).

$$\ell(x, \delta, t) = \ell_{model}(f(x + \delta), t) \\ + c \cdot \ell_{metric}(x, x + \delta) \tag{1}$$

The specific steps to implement audio injection attack is as follow:

*Table 1.* Commands and action records with Google Assistant.

| *ok google play some music* and *hello world* | a) My pixel started play music |
| | b) It answered "Now you're speaking my language" |
| *ok google airplane mode off* and *that's cool* | a) My pixel turn off the airplane |
| | b) It answered "You got it" |
| *ok google read mail* and *I love you* | a) My pixel listed my latest email |
| | b) It answered "That's high praise coming from you" |

1. Use ASL module to locate the blank region at the front of the original audio. For example, shown in Figure 1, the input of ASL are original sequence $\pi$ and our target "OPEN THE DOOR AND ", then it will output all the locations of different characters. Finally, we only use the foremost or rearmost positions (first 3 in the example).

2. Repeat gradient descent calculation on these points based on the normal loss function $l(\cdot)$ in Formula 1, until get the adversarial example.

Assuming that the original semantic information of audio is $p$, we set the target phrase $t$ as "*open the door and* " (not including $p$). Our audio injection attack will attack each audio and make them transcribed as "*open the door and* " + $p$. Experimental results show that audio injection attack generated adversarial examples with a success rate of 85%. Most of their $dB_x(\theta)$ are greater than 40dB, which means the distortion is very slight and it almost unnoticeable to humans. More importantly, even if the added perturbation is very large, the audio vector part containing the original semantic information has not been modified because the perturbation is only added on a small section of the audio head, which makes it still unnoticeable to humans.

In addition, considering the phonetic correlation of the context in the predicted sequence, we suggest that only the first 70% points in the injected part should be perturbed to avoid obvious influence on the following parts.

An interesting question is what if we directly set the target phrase as "*open the door and* " + $p$, instead of perturbing the beginning of the audio. Then we carried out the experiment with Carlini & Wagner's approach(Carlini & Wagner, 2018).

In order to show the difference in results more clearly, we randomly select three adversarial examples and visualize their spectrograms in Figure 2. Compared with the spectrogram of the original audio, it can be clearly seen that the sample generated by our method has only slight distortion to the front part (red rectangular area), while C&W's attack has distortion almost on every point of the audio (black rectangular area).

We further notice that after the same number of iterations, the adversarial examples generated by C&W's method often leads to context confusion, i.e., overlapping words or lost words. For example, when $t$ is "*it was dropping*", the final adversarial example will be "*open the donit was dropping*". While our method can generate adversarial examples transcribed as $t + p$ without confusion perfectly.

Actually, if the victim doesn't notice the initial slight distortion, the victim will be even more unaware of the adversarial audio because the rest of the whole audio is exactly the same as the original audio. This feature once again enhances the concealment of our attack.

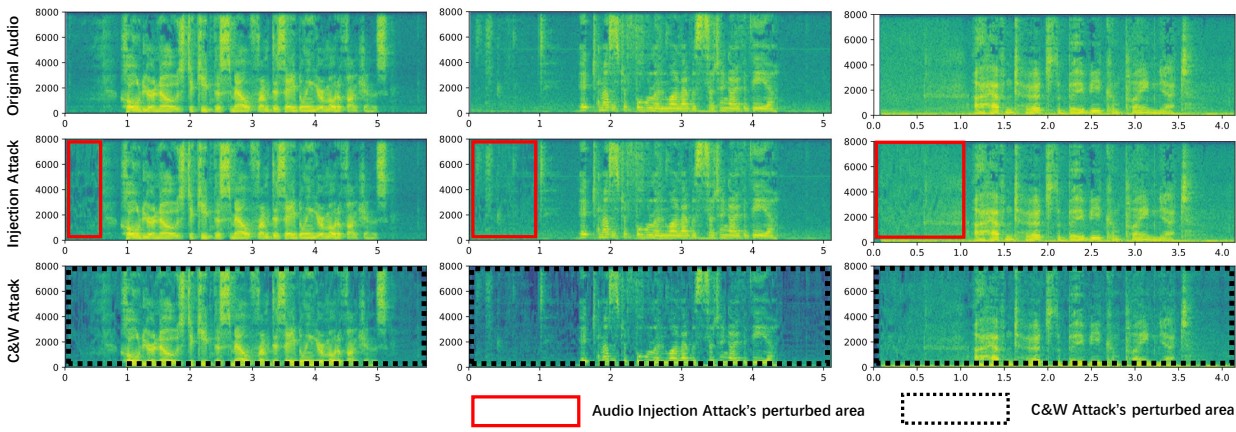

*Figure 2.* Spectrograms of audio adversarial examples generated by Audio Injection Attack and C&W's attack.

## 4. Notations and Definitions

All notations and definitions used in our paper are listed in Table 2.

*Table 2.* Notations and Definitions used in our paper.

| | |
|---|---|
| $x$ | The original input audio |
| $\delta$ | The distortion to the original audio |
| $p$ | The phrase of the semantic information of original audio |
| $t$ | The targeted texts |
| $f(\cdot)$ | The threat model |
| $\ell(\cdot)$ | The loss function to generate audio adversarial examples |
| $\ell_{model}(\cdot)$ | The loss function to measure the difference between the current output of the model and the targeted texts |
| $\ell_{metric}(\cdot)$ | The loss function to limit the difference between the adversarial examples and the original samples |

## 5. CONCLUSION

This paper proposes the audio injection adversarial example attack. To best of our knowledge, We are the first in the field to both take the factors of the attack success rate and the attack concealment into consideration during the generation of audio adversarial examples. Our experimental results show that our method can increase the concealment of attack behavior in actual attack scenario. Our future work will focus on the defense to audio adversarial examples.

## Acknowledgements

This work was partially supported by the Director of Computer Application Research Institute Foundation SJ2020A08 and China Academy of Engineering Physics Innovation and Development Fund Cultivation Project PY20210160.

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
