# OpenReview forum: "Audio Injection Adversarial Example Attack"
_ICML.cc/2021/Workshop/AML — ICML 2021 Workshop AML Poster_

### Official Review · Reviewer_moCW · 2021-06-20
**A reasonable applying situation**

**Rating:** Accept
**Confidence:** 5

**Review:**

This paper proposed an audio injection adversarial attack to make an automatic speech recognition (ASR) system to additionally output target phrases in front of its original outputs. It takes effect by adding perturbations at the foremost or rearmost positions of the original audio sequence. Comparing to existing methods, it can have better concealment because the victim will still get his desired instruction execution result. In addition, it uses fewer perturbations to implement attacks.
ASR)

Strengths:
1.	This method applies to a reasonable situation that there could be possible detections to check if the desired results are satisfying.

Weaknesses:
1.	The comparison result of the proposed method and existing methods lacks quantitative analysis both in attack success rate and perturbation magnitude.
2.	The authors did not clarify the ASR system they used. It is also recommended to compare different models as well as doing a transfer study.

---

### Decision · Program_Chairs · 2021-06-21

**Decision:**

Accept (Poster)

**Comment:**

This paper proposed an audio injection adversarial attack to make an automatic speech recognition (ASR) system to additionally output target phrases in front of its original outputs.  The authors can further address the reviewer's comments.